# N-Terminal Pro-B Type Natriuretic Peptide as a Predictive Biomarker of Bronchopulmonary Dysplasia or Death Due to Bronchopulmonary Dysplasia in Preterm Neonates: A Systematic Review and Meta-Analysis

**DOI:** 10.3390/jpm13091287

**Published:** 2023-08-23

**Authors:** Kalliopi Rodolaki, Vasilios Pergialiotis, Ioakeim Sapantzoglou, Marianna Theodora, Panagiotis Antsaklis, Kalliopi Pappa, Georgios Daskalakis, Aggeliki Papapanagiotou

**Affiliations:** 11st Department of Pediatrics, National and Kapodistrian University of Athens, Aghia Sophia Children’s Hospital, 11527 Athens, Greece; kelli1_@hotmail.com; 21st Department of Obstetrics and Gynecology, Alexandra Hospital, National and Kapodistrian University of Athens, 11527 Athens, Greece; pergialiotis@yahoo.com (V.P.); kimsap1990@hotmail.com (I.S.); martheodr@gmail.com (M.T.); panosant@gmail.com (P.A.); kalliopi.pappa20@gmail.com (K.P.); gdaskalakis@yahoo.com (G.D.); 3Department of Biological Chemistry, School of Medicine, National and Kapodistrian University of Athens, 11527 Athens, Greece

**Keywords:** NT-proBNP, bronchopulmonary dysplasia, preterm birth, premature neonate, meta-analysis

## Abstract

Background: Emerging evidence suggests the clinical utility of N terminal pro B type natriuretic peptide (NT-proBNP) in multiple cardiac and pulmonary abnormalities both in adult and pediatric populations. To date, however, there is no consensus regarding its efficacy for the prediction and severity of bronchopulmonary dysplasia in premature neonates. The objective of the present meta-analysis was to determine differences in NT-proBNP among neonates that develop BPD or die from BPD and to evaluate if there is relative information on the diagnostic accuracy of the method. Methods: We conducted a systematic search according to the PRISMA guidelines and looked into Medline (1966–2023), Scopus (2004–2023), Clinicaltrials.gov (2008–2023), EMBASE (1980–2023), Cochrane Central Register of Controlled Trials CENTRAL (1999–2022) and Google Scholar (2004–2023) together with the reference lists from included studies. The potential risk of bias encountered in our study was evaluated using the QUADAS -2 tool. Finally, a total of 9 studies met the eligibility criteria, comprising 1319 newborns, from which 397 developed BPD and 922 were unaffected controls. Results: The results retrieved from our meta-analysis showed that newborns suffering from BPD had notably elevated NT-proBNP levels after birth when compared with healthy neonates (SMD 2.57, 95% CI 0.41, 4.72). The summary effect of the AUC meta-analysis showed that NT-proBNP was very accurate in detecting neonates at risk of developing severe BPD or dying from the disease (AUC −0.16, 95% CI −0.23, −0.08). No studies reported data relevant to the sensitivity and/or specificity of the method in diagnosing BPD. Conclusion: Serum NT-proBNP levels represent a potential future biomarker with great diagnostic validity for the prediction of BPD complicating preterm deliveries. The limited amount of studies included and the significant variations in cutoff values and timing of measurement still restrict the application of NT-proBNP as an established clinical biomarker for BPD. The design of larger prospective studies will provide a more representative number of participants and will address the discrepancies in existing literature.

## 1. Introduction

Bronchopulmonary dysplasia (BPD) is a chronic lung disorder that represents one of the most frequent and severe sequelae complicating premature deliveries. Despite the great advances in perinatal care, the prevalence of bronchopulmonary dysplasia among preterm babies remains constant, possibly due to the rising survival rates of newborns born prematurely [1]. The complex pathophysiology of BPD mainly lies in an exaggerated inflammatory response and lung immaturity, which leads to defective repair processes and creates anatomical lung alterations like pulmonary vascular dysgenesis and disrupted alveolarization. Unfortunately, neonates with BPD encounter serious neonatal morbidity and mortality, while survivors face great risk for future neurodevelopmental impairment and vulnerability to progressive lung function decline and cardiac dysfunction [2]. Multiple factors seem to contribute to the progression to BPD, with the most important being low gestational age and birth weight, the use of mechanical ventilation after birth with induced lung injury, oxygen toxicity, male sex, tracheal colonization with ureaplasma, epigenetics, maternal chorioamnionitis, patent ductus arteriosus (PDA) and late-onset sepsis [3]. Taking into consideration the serious short and long-term consequences of BPD, it is of utmost importance to identify candidate biomarkers that could predict the development of BPD in high-risk neonates in order to implement preventative measures for modifiable variables and targeted therapeutic approaches.

B-type natriuretic peptide (BNP) and its inactive derivative N terminal pro B type natriuretic peptide (NT-proBNP) are secreted by ventricular cardiomyocytes after the trigger of pressure or ventricular blood volume overload, which leads to the stretching of the myocardial wall. NT-proBNP appears to be more stable in serum samples for at least 72 h and has a longer half-time (60–120 min) in circulation, which is why it is considered a more sensitive and accurate biomarker, as it may reflect higher and more consistent concentrations in the circulation. Although the clinical utility of natriuretic peptides has been extensively studied in the adult population, their prognostic role for pediatric and neonatal disease is still under evolution [4]. Indeed, according to several recent studies, NT-proBNP has shown great diagnostic performance for the prediction of pulmonary hypertension and hemodynamically significant patent ductus arteriosus in preterm infants as well as for cardiac abnormalities, Kawasaki syndrome and MISC in older children [5,6,7,8,9].

Taking all this data into account, we systematically searched the international literature in order to evaluate the possible prognostic role of systemic NT-proBNP levels after birth for the development of bronchopulmonary dysplasia and death in preterm infants. 

## 2. Methods

### 2.1. Protocol and Registration

We designed this meta-analysis using the guidance of the Preferred Reporting Items for Systematic Reviews and Meta-Analyses (PRISMA) guidelines [10]. Only aggregated data were used during the analysis that were already published in the international literature; therefore, patient consent and institutional review board approval were not retrieved as they are not required in this type of study. The study’s protocol was published in PROSPERO (International Prospective Register of Systematic Reviews) prior to the conduct of this review (Registration number: CRD42023387826).

### 2.2. Eligibility Criteria

The criteria that determined eligibility for inclusion in studies were predetermined. Observational studies that compared differences in the mean expression of NT-proBNP among neonates with BPD and controls, as well as those that reported the diagnostic accuracy of the method in detecting BPD, were considered eligible for inclusion. Only studies published in the English language were included in our meta-analysis. The timing of measurement of the NT-proBNP was evaluated and reported on the grounds of performing meta-regression analysis.

### 2.3. Information Sources and Search Strategy

Two authors (K.R. and I.S.) searched Medline (1966–2022), Scopus (2004–2022), Clinicaltrials.gov (2008–2023), EMBASE (1980–2023), Cochrane Central Register of Controlled Trials CENTRAL (1999–2023) and Google Scholar (2004–2023) along with the reference lists of electronically retrieved full-text papers. The date of the last search was set at 10 June 2023. The search strategy included the text words “NT-proBNP; natriuretic peptide; bronchopulmonary; dysplasia; preterm neonate” and is presented in brief in Figure 1.

The process of study selection was completed in three consecutive stages. Firstly, we deduplicated the retrieved articles, and following that, two independent reviewers screened the remaining articles (K.R., I.S.) and assessed their eligibility. The definitive decision for inclusion was taken only after retrieving and reviewing the full version of articles that were considered potentially eligible. In the case of discrepant opinions, a consensus was requested by all authors prior to the inclusion of the study in the meta-analysis.

### 2.4. Study Selection and Data Extraction

The outcome measures were predetermined at the phase of design of the present systematic review. Cochrane’s data collection form for intervention reviews for RCTs and non-RCTs was used to perform data extraction [11].

The main outcomes of our study were the difference in the NT-proBNP levels among neonates that developed BPD or death and controls and the produced area under the curve (AUC) of the diagnostic accuracy of the method.

### 2.5. Assessment of Risk of Bias and Quality of Evidence

The QUADAS-2 (Quality Assessment of Diagnostic Accuracy Studies-2) tool was used to evaluate the methodological quality of included studies. The tool evaluates each study, including the processes that are used for patient selection, the indicated indexed test, the reference standard to that test and the flow and timing of the test/study.

The Grading of Recommendations Assessment, Development and Evaluation (GRADE) framework was used to evaluate the quality of the evidence. The range of quality ranges from very low to high. A summary of the studies’ limitations, directness, consistency, precision and publication bias is constructed, and the credibility of the evidence is constructed considering these variables. 

### 2.6. Synthesis of Results

The meta-analysis was performed with RStudio using the *meta and metafor* functions (RStudio Team (2015). RStudio: Integrated Development for R. RStudio, Inc., Boston, MA, USA, URL http://www.rstudio.com/ accessed on 1 June 2023.). Taking in mind the anticipated methodological heterogeneity of included studies, we did not consider the statistical heterogeneity for the evaluation of the appropriate model of statistical analysis as the assumption of comparable effect sizes was considered violated by definition [12]. Confidence intervals were set at 95%. The pooled summary effect sizes (standardized mean differences) (SMD) and areas under the curve (AUC) of NT-proBNP for the detection of BPD or death from BPD, as well as the retrieved 95% confidence intervals (CI) were calculated with the Hartung-Knapp–Sidik–Jonkman instead of the traditional Dersimonian-Laird random effects model analysis (REM). This decision was based on the ability of the former model to provide more credible summary effect estimates when the sample size and the heterogeneity of included studies are considerable [13]. The transformation of variables that were expressed as mean and 95% confidence interval values to the log mean and log standard deviation values was performed to render the analysis of pre-calculated summary effect estimates. 

Publication bias was assessed using graphical methods (funnel plots) as well as by evaluating the possibility of small-study effects. Egger’s regression and Begg–Mazumdar’s rank correlation tests were used to evaluate the asymmetry of funnel plots. The Duval and Tweedie’s trim and fill test was applied to evaluate the possibility of variability of findings in the case of asymmetrical funnel plots. A minimum of 10 studies was considered as a prerequisite to apply this test. 

To exclude the possibility of publication bias due to the p-hacking limit, a meta-analysis using Rucker’s model was performed. Outlier analysis was also performed to evaluate the independent effect of studies that provided extremely different data to that of the aggregated model. 

### 2.7. Prediction Intervals

Prediction intervals (PI) were also calculated, using the meta function in RStudio, to evaluate the possibility of retrieving similar results in future studies. The test takes into account the inter-study variation of the results and expresses the existing heterogeneity at the same scale as the examined outcome. 

### 2.8. Meta-Regression Analysis

Meta-regression analysis was planned to evaluate the effect of time on the results of the primary analysis. When studies referred to a timeframe of collection of NT-proBNP rather than an accurate day in the life of the neonate, the median day of this interval was used in the meta-regression analysis.

## 3. Results

Nine studies were selected for inclusion in the present meta-analysis [14,15,16,17,18,19,20,21,22]. Significant discrepancies were observed in the methodological characteristics of the included studies, which are indicated in Table 1. Specifically, while all of them excluded neonates that had congenital heart abnormalities and major anomalies, others included more strict criteria. Moreover, the timing of assessment of NT-proBNP significantly varied as some studies evaluated differences during the first days of life, whereas others expanded this further during the end of the first month or even later. In Table 2 we summarized differences among cases that developed BPD or death and control neonates for the most important investigated factors. The methodological quality of included studies revealed significant methodological issues, primarily in the domains of patient selection and index test, which was severely downgraded due to the case–control methodology of included studies (Figure 2).

The analysis of standardized mean differences indicated that neonates that developed BPD had significantly higher values of NT-proBNP compared to preterm control neonates (SMD 2.57, 95% CI 0.41, 4.72, Figure 3). Statistical heterogeneity was particularly high (I-square test = 97%). The fixed effect model analysis revealed that the difference remained significant with a somehow smaller summary effect estimate (SMD 1.56, 95% CI 1.41, 1.71). Prediction intervals indicated, however, that future studies might not support a significant difference among the two groups. This might indicate the presence of potential outliers that might influence the results of the primary analysis. Taking this into account, we performed a sensitivity analysis, which revealed that following the exclusion of statistically detected outliers, the effect remained statistically significant, with a lower summary effect estimate but more precise confidence intervals (SMD 1.31, 95% CI 1.16, 1.46). The trim and fill analysis was also used to help decrease the bias that was detected in the Contour-enhanced Funnel plot. However, the analysis did not result in different summary effect estimates compared to those of the primary analysis. The p-curve analysis revealed that the summary effect estimate had an evidential value, thus excluding the possibility of p-hacking. Meta-regression analysis indicated that significant residual heterogeneity remained after considering the timing of NT-proBNP collection (I-square 99%, *p* < 0.001) and that this variable could only explain 10% of the variability in outcome reporting and its effect could not significantly impact the results of the primary analysis (estimate −0.12, 95% CI −0.32, 0.09, *p =* 0.209). Subgroup analysis considering neonates that developed BPD only revealed that the SMD did not differ significantly compared to those that did not develop (SMD 1.01, 95% CI −0.42, 2.44). Evaluating the studies that considered neonates that developed moderate-severe BPD or died from BPD, we observed that differences marginally reached the statistical significance (SMD 3.34, 95% CI −0.03, 6.72). (Figure 4).

The analysis of AUC effect estimates indicated that the estimation of NT-proBNP was very accurate in detecting neonates at risk of developing severe BPD or even dying from the disease (AUC −0.16, 95% CI −0.23, −0.08, Figure 5). Similar results were obtained from the fixed effects analysis (AUC −0.13, 95% CI −0.15, −0.10), even though statistical heterogeneity was high (I-square test = 77%). Prediction intervals indicated that the effect was close to, but did not reach statistical significance, therefore leaving questions about the results of future studies. Outlier analysis did not reveal the presence of studies that significantly deviated from the mean observed estimate. Three studies were added with the trim and fill analysis, indicating that publication bias might be an issue; however, even after their inclusion, the summary AUC remained significant (AUC −0.11, 95% CI −0.19, −0.03, *p =* 0.010). P-curve analysis indicated the presence of an evidential value, therefore excluding the possibility of p-hacking. Meta-regression analysis revealed that the timing of collection could explain 28.7% of the heterogeneity observed in the primary analysis; however, this sufficed to reduce the residual heterogeneity to non-significant (*p =* 0.13). However, the actual effect of the timing of collection on the actual summary effect estimate was not significant (estimate 0.01, 95% CI −0.003, 0.02, *p =* 0.10). Only one study included BPD as a solitary outcome, and its results did not reveal a significant prognostic accuracy of NTproBNP (AUC −0.04, 95% CI −0.08, 0.01) (Figure 6). The summary effect of studies involving moderate-severe BPD or death as an outcome was significant (AUC −0.17, 95% CI −0.21, −0.12). 

## 4. Discussion

To the best of our knowledge, this is the first systematic review and meta-analysis in the current literature that addresses the association between Nt-proBNP levels after birth and the subsequent development of BPD in preterm neonates by including all the relevant published data. In alliance with our original hypothesis, NT-proBNP levels were found markedly elevated in preterm neonates developing BPD, and the hormone demonstrated a substantial predictive value for BPD progression or death. Unfortunately, not all included studies were homologous in their methodology. Therefore, an additional subgroup analysis was conducted to assess differences in different outcomes, one being the development of BPD only and another being the development of moderate to severe BPD or death. All of the studies included in our meta-analysis reported a strong association between elevated NT-proBNP levels after birth and the occurrence of BPD in preterm infants, although there was no accordance concerning the postnatal day of blood collection. 

NT-proBNP is known to be a clinically measurable natriuretic neurohormone secreted by the ventricular musculature in response to volume or pressure overload, which demonstrates higher levels in patients with cardiac disease. Increasing evidence supports the utilization of the aforementioned biomarker in screening, diagnosing and management of children with a number of heart defects [23]. As such, the studies that were analyzed excluded infants with severe congenital heart defects, given the fact that their presence would be an obvious reason for increased NT-proBNP levels. According to the literature, pulmonary entities that would eventually cause raised NT-proBNP levels due to the underlying cardiac stress include the persistent pulmonary hypertension of the newborn (PPHN) and Acute Respiratory Distress Syndrome (ARDS), and several studies have associated the high NT-proBNP levels with ventilator weaning and extubation failure [24]. However, in our sample of preterm infants, a pulmonary cause was more likely compared to an underlying cardiac one, with several mechanisms possibly explaining such a rise, such as the increased pulmonary vascular pressure and chronic lung injury [25]. 

It has been strongly suggested that various complications of prematurity may affect NT-proBNP levels, namely hemodynamically significant patent ductus arteriosus (HsPDA), pulmonary hypertension, BPD, retinopathy of prematurity, inflammation, sepsis and possibly necrotizing enterocolitis and intraventricular haemorrhage (IVH) [26]. Bronchopulmonary dysplasia is considered a neonatal form of chronic inflammatory pulmonary disorder with characteristic clinical and radiographic presentation, affecting newborns born prematurely. BPD represents a clinical diagnosis that was classically defined by the need for oxygen supplementation or respiratory support at 36 weeks corrected gestational age and classified as no, mild, moderate or severe according to the National Institute of Health (NIH) consensus in 2001. However, this definition has evolved with time and has raised great disputes among researchers [27]. From a pathophysiological point of view, BPD appears to result from a combination of endothelial injury, vascular growth disruption, barotrauma and alveolar influx of numerous proinflammatory cytokines. In the long term, children with a history of BPD after birth run the risk of having impaired pulmonary and cardiac function, susceptibility to infections and poor neurodevelopmental outcomes due to the extended use of corticosteroids. Therefore, considering the detrimental effects of BPD from the neonatal period throughout childhood, previous researchers have attempted to propose several potential biomarkers that may be applied as risk stratification tools for the early diagnosis and management of BPD, although none of them is routinely used by physicians yet [28]. Preterm infants who are at greater risk for moderate to severe BPD may benefit from the early administration of corticosteroid therapy and avoid the detrimental sequelae of BPD. BPD subjects have been associated with evidence of echocardiographic abnormalities related to right ventricular diastolic dysfunction, elevated right atrial pressure and volume overload, which justifies the assumption that NT-proBNP levels are correlated with BPD progression [29]. 

PDA is one of the most commonly encountered sequelae of prematurity in neonates born before 32 weeks of gestation. Unarguably, a large left-to-right PDA shunt has been widely linked with BPD development before the postmenstrual age of 36 weeks. Indeed, the results of the included studies in our meta-analysis indicated a higher incidence of BPD development in neonates with established HsPDA. PDA results in lung compliance changes and excessive pulmonary blood flow, which may obstruct the maturity of the developing lungs and predispose affected neonates to chronic lung disease. Elevated levels of NT-proBNP have been identified in newborns with PDA, possibly due to the fact that the increased pulmonary flow created in the presence of a PDA results in left heart volume overload, which directly increases the secretion of natriuretic peptides [30]. Thus, the presence of a HsPDA may significantly confound the relationship between Nt-proBNP levels and BPD progression and should be taken into account carefully. However, strong evidence derived from the majority of our included studies demonstrates that NT-proBNP may serve as a great diagnostic tool for BPD regardless of the presence of a PDA. Joseph et al., in their study, performed an echocardiogram in all neonates with BPD in order to exclude the possibility of concomitant cardiovascular or pulmonary disorders like PDA or PPHN that could also provoke a rise in NT-proBNP. These authors were the first to demonstrate higher levels of NT pro-BNP in preterm infants affected by BPD compared to controls at 28 days of life and, secondly, increased levels of NT-proBNP in healthy preterm newborns compared to the reported values of normal-term infants, suggesting that there is an underlying degree of lung damage even in premature infants that do not fulfil the BPD criteria [14]. Semmler et al. confirmed the acceptable predictive value of raised NT-proBNP levels at day 3 postnatally in the development of BPD, and these authors were the first to associate those levels with death, with the mortality frequency increasing as the levels move from the lowest to the highest quartiles. Although this association remained present in neonates without a clinically significant patent ductus arteriosus (PDA), it was found stronger when PDA coexisted [15]. Blanco et al. prospectively addressed the diagnostic accuracy of NT-proBNP both during the first 48–96 h of life and later at 5–10 days of life. However, their results indicated that a strong correlation between NT-proBNP and BPD development unaffected by the appearance of HsPDA is present only at 5–10 days of life with a sensitivity of 82% and specificity of 83% [17]. Similarly, the study of Montaner et al. revealed that premature babies with NT-proBNP levels above 17,800 pg/mL at day 2–3 had a higher tendency towards BPD development independently from the presence of a PDA [16]. Furthermore, the study from Potchiurko et al. showed that NT-proBNP levels ≥ 3537 pg/mL at 8–9 days of life have a great diagnostic capacity for BPD progression independently from PDA persistence. 

Song et al. assessed the potential association between NT-proBNP levels with moderate-to-severe BPD or death, demonstrating that, even after adjusting for potential confounding factors, a cutoff value of 3360 ng/L on day 7 showed a quite promising predicting ability with its sensitivity and specificity being 80% and 86.2%, respectively [22]. In their retrospective cohort study, Zhou Lin et al. studied 147 preterm infants and revealed that a serum NT-proBNP cutoff value of 2002.5 pg/mL at the first day of life has a sensitivity of 87.5% and specificity of 74.7% to predict moderate/severe BPD or death [20].

Mendez-Abad et al. examined the potential predictive value of NT-proBNP in preterm infants that will eventually develop BPD, revealing that the assessment of the biomarker at 14 days of life had an even better performance with a sensitivity of 100% and a specificity of 86% [19]. As indicated by a recent case–control study, which was excluded from our meta-analysis, in accordance with the results of Mendez -Abad et al., NT-proBNP levels had the strongest diagnostic accuracy for BPD progression on the 14th of life while on the 7th day of life, its predictive capacity was significantly lower. Moreover, the same study revealed that lung ultrasound score (LUS), which has been extensively proposed as a non-invasive alternative for the detection of numerous neonatal respiratory disorders, had greater diagnostic ability than NT-proBNP for BPD prediction on the 7th day of life [31]. 

## 5. Strengths and Limitations

The main strength of our study is the inclusion of all the available published data on the issue of the association of the levels of NT-proBNP with the development of BPD. Our findings resulted from the analysis of the data from 1319 newborns (397 cases and 922 controls), and there was a clear agreement among the included studies on the definition of the outcome under examination. Furthermore, similar exclusion criteria were applied among the various studies with newborns with chromosomal/genetic abnormalities and major heart or anatomical anomalies not included in the analysis performed. Additionally, in an effort to reduce heterogenicity due to the different outcomes under question in the included studies, a subgroup analysis was performed to address the issue of the association of the levels of NT-proBNP with either the development of BPD only or the occurrence of moderate to severe BPD or death from BPD. 

Nevertheless, we need to acknowledge that our meta-analysis has faced several limitations. Our main limitation is the methodological and clinical heterogenicity of the various included studies. Three of the included studies were retrospective cohorts, 2 of them were case–control studies and four of them were prospective cohorts in design. While the outcome under question is similarly defined among the included studies, the timing of the blood collection of NT-proBNP varies significantly. To be more specific, two study groups assessed the NT-proBNP at 28 days of life, two others at 2–3 days of life, 1 study group between 5–10 days and the last one within the first 24 h. To continue, three studies performed a consecutive assessment of the levels with blood collections on several different days in an effort to establish the optimal time period for the prediction of BPD. As such, the study designs differ from each other, and while the provided results are of importance, further studies should be designed for concrete conclusions to be extracted. Specifically, large-scale prospective cohort studies with sequential NT-proBNP measurements during the first month of life are needed in order to detect the optimal day of life that would maximize the diagnostic performance of the biomarker for later development of BPD. Moreover, the threshold cutoff points of NT-proBNP levels differ significantly among the identified studies, possibly due to the different postnatal days of NT-proBNP measurement. It has been suggested that NT-proBNP levels are independently influenced by gestational age and postnatal chronological age in preterm infants. Indeed, normative NT-proBNP levels are higher during the first week of life and gradually decline towards the end of the first month of life to reach a stable plateau [32]. Another important limitation of our study is the exclusion of studies relevant to our subject after reading the full text because they did not provide details about NT-proBNP levels and could not be incorporated into our meta-analysis. Lastly, the studies included in the present systematic review did not report data relevant to the diagnostic accuracy (sensitivity, specificity) of NT-proBNP on the diagnosis of BPD; hence, further research is needed to establish definitive cutoff values and investigate the diagnostic accuracy of the method.

## 6. Conclusions

The outcomes of our present meta-analysis highlighted that serum NT-proBNP levels display a promising diagnostic performance for the development of severe BPD or death in premature neonates. However, despite the strong indications for the usefulness of NT-proBNP, its diagnostic accuracy must be more thoroughly examined with future well-designed prospective studies with a multicenter character. Risk stratification of neonates with BPD is crucial for the proper management and optimal clinical outcomes with less severe sequelae. Early patient categorization according to specific Nt-proBNP levels may contribute to the identification of patients at greater risk for poorer outcomes and the implementation of individualized treatment strategies. The conduction of larger-scale studies will hopefully minimize the heterogeneous data and, consequently, the risk of bias presented in the current literature. Our study may serve as a pilot one for other researchers investigating the field in order to define the optimal timing of serum NT-proBNP measurement as well as establish an ideal cutoff value that will be generally acknowledged and incorporated in clinical practice. 

## Figures and Tables

**Figure 1 jpm-13-01287-f001:**
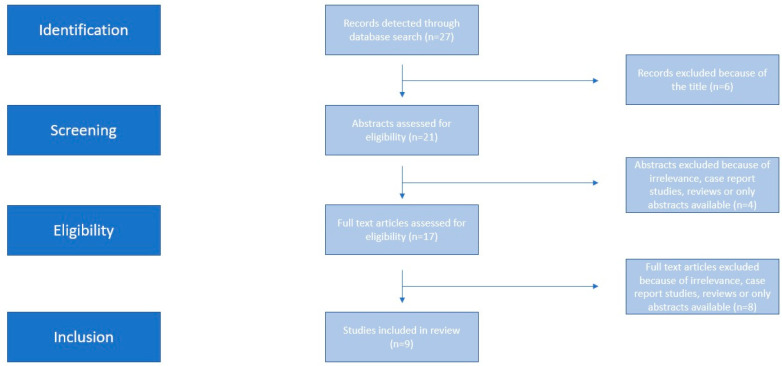
Search strategy.

**Figure 2 jpm-13-01287-f002:**
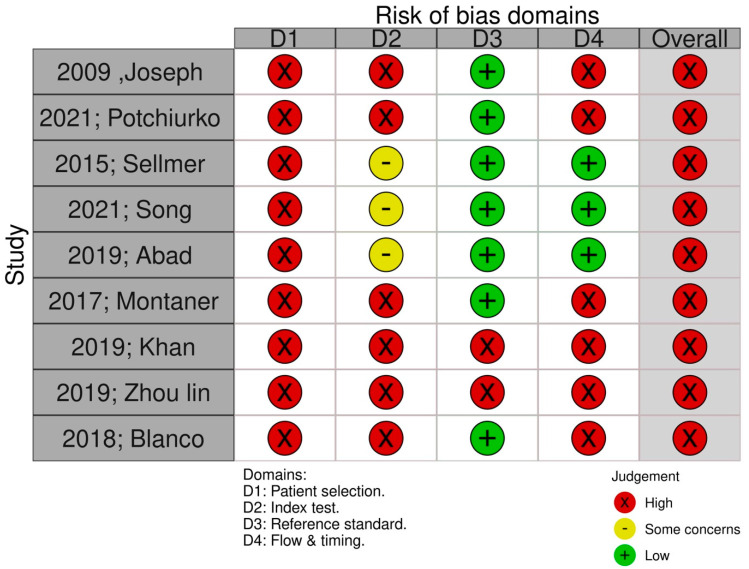
Methodological evaluation of included studies with the QUADAS-2 tool. 2009, Joseph [14]; 2021, Potchiurko [21]; 2015, Sellmer [15]; 2021, Song [22]; 2019, Abad [19]; 2017, Montaner [16]; 2019, Khan [18]; 2019, Zhou Lin [20]; 2018, Blanco [17].

**Figure 3 jpm-13-01287-f003:**
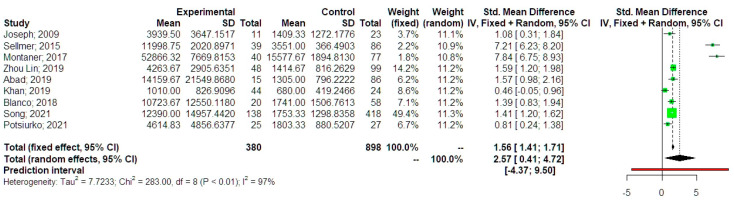
Mean differences of serum NT-proBNP levels between neonates with bronchopulmonary dysplasia or death and healthy controls. Forest plot analysis: Vertical line = “no difference” point between the two groups. Green squares = mean differences; Diamond = pooled mean differences and 95% CI for all studies; Horizontal black lines = 95% CI; Horizontal red line = prediction intervals. Results from the Mantel-Haenszel fixed effects and the Sidik–Jonkman random effects model are presented. 2009, Joseph [14]; 2015, Sellmer [15]; 2017, Montaner [16]; 2019, Zhou Lin [20]; 2019, Abad [19]; 2019, Khan [18]; 2018, Blanco [17]; 2021, Song [22]; 2021, Potchiurko [21].

**Figure 4 jpm-13-01287-f004:**
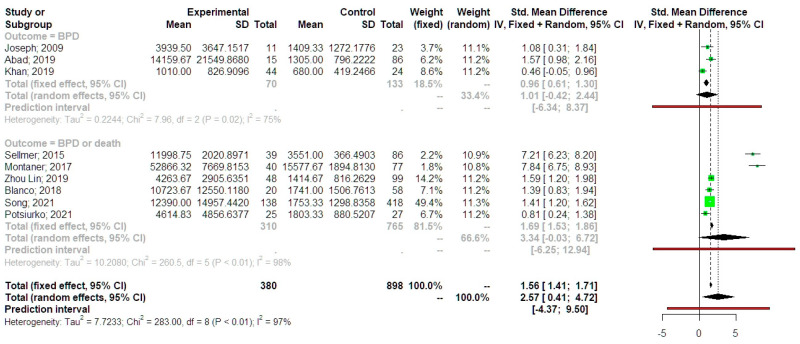
Subgroup analysis of studies included neonates who developed only BPD and those who developed severe BPD or death. Forest plot analysis: Vertical line = “no difference” point between the two groups. Green squares = mean differences; Diamond = pooled mean differences and 95% CI for all studies; Horizontal black lines = 95% CI; Horizontal red line = prediction intervals. Results from the Mantel-Haenszel fixed effects and the Sidik–Jonkman random effects model are presented. 2009, Joseph [14]; 2019, Abad [19]; 2019, Khan [18]; 2015, Sellmer [15]; 2017, Montaner [16]; 2019, Zhou Lin [20]; 2018, Blanco [17]; 2021, Song [22]; 2021, Potchiurko [21].

**Figure 5 jpm-13-01287-f005:**
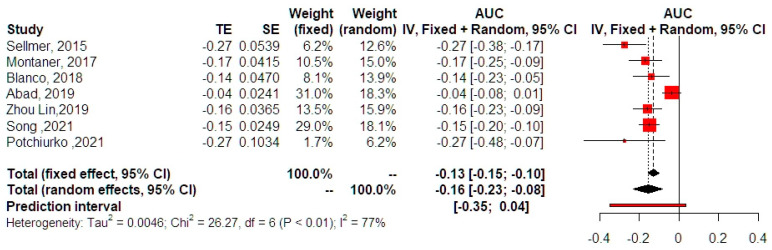
Summary effect of AUC meta-analysis. Forest plot analysis: Vertical line = “no difference” point between the two groups. Red squares = AUC; Diamond = pooled AUC and 95% CI for all studies; Horizontal black lines = 95% CI; Horizontal red line = prediction intervals. Results from the Mantel-Haenszel fixed effects and the Sidik–Jonkman random effects model are presented. Continuous small dotted vertical line = pooled mean AUC according to the REM, continuous large dotted vertical line = pooled mean AUC according to the FEM. 2015, Sellmer [15]; 2017, Montaner [16]; 2018, Blanco [17]; 2019, Abad [19]; 2019, Zhou Lin [20]; 2021, Song [22]; 2021, Potchiurko [21].

**Figure 6 jpm-13-01287-f006:**
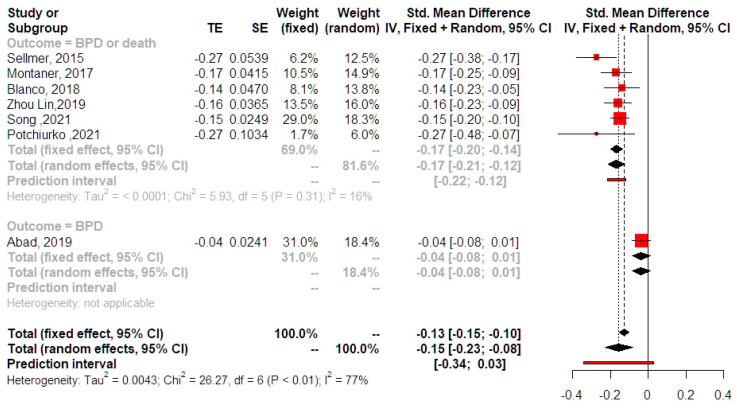
Summary effect of AUC meta-analysis by subgroups. Forest plot analysis: Vertical line = “no difference” point between the two groups. Red squares = AUC; Diamond = pooled AUC and 95% CI for all studies; Horizontal black lines = 95% CI; Horizontal red line = prediction intervals. Results from the Mantel-Haenszel fixed effects and the Sidik–Jonkman random effects model are presented. Continuous small dotted vertical line = pooled mean AUC according to the REM, continuous large dotted vertical line = pooled mean AUC according to the FEM. 2015, Sellmer [15]; 2017, Montaner [16]; 2018, Blanco [17]; 2019, Zhou Lin [20]; 2021, Song [22]; 2021, Potchiurko [21]; 2019, Abad [19].

**Table 1 jpm-13-01287-t001:** Methodological characteristics of included studies.

Year, Author	Country	Type of Study	Exclusion Criteria	Type of Sample	Timing of Collection
2009, Joseph [14]	Israel	Prospective cohort	Major anomalies Congenital heart defects Current sepsis Current PDA Lack of informed consent of parents/guardians	Serum	28 days of life
2015, Sellmer [15]	Denmark	Prospective cohort	Chromosomal abnormalities Congenital heart defects (except for ASD) Lack of informed consent of parents/guardians	Serum	3 days of life
2017, Montaner [16]	Spain	Retrospective cohort	Chromosomal disorders and other malformations, Cardiac disease causing circulatory overload other than PDA, Incomplete echocardiographic and biochemical evaluation	Serum	2 to 3 days of life
2018, Blanco [17]	Spain	Prospective cohort	Major congenital defects, death during first 48 h, incomplete data, Lack of informed consent of parents/guardians	Serum	5–10 days of life
2019, Khan [18]	USA	Prospective	Congenital anomalies, sepsis, perinatal asphyxia, complex congenital heart disease and PDA	Serum	28 days of life
2019, Abad [19]	Spain	Prospective cohort	Congenital heart defects (except for PFO, PDA, ASD, VSD < 2 mm) Genetic syndrome Major anomalies Death in 1st week of life Lack of informed consent of parents/guardians	Serum	1, 3, 7, 14, 28, 35, 42, and 49 days after birth
2019, Zhou Lin [20]	China	Retrospective cohort	Genetic disorders, congenital anomalies including complex congenital heart disease, death on the first day of life	Serum	1st day of life
2021, Potchiurko [21]	Ukraine	Retrospective cohort	Congenital heart defects (except for PDA) Haemorrhaging syndrome Necrotizing enterocolitis Lung hypoplasia Platelets < 50,000/mm^3^ Oliguria < 1 mL/kg/h Lack of informed consent of parents/guardians	Serum	2, 3, 8, 9 days of life
2021, Song [22]	China	Retrospective cohort	Congenital metabolic defects Congenital heart defects(except for PFO and PDA) Severe renal insufficiency Death in 1st week of life Incomplete data Lack of informed consent of parents/guardians	Serum	1, 3, 7, 14, 21, 28 days of life

Abbreviations: PDA, Patent ductus arteriosus; PFO, Patent foramen ovale; ASD, atrial septal defect; VSD, ventricular septal defect; IVH, intraventricular haemorrhage

**Table 2 jpm-13-01287-t002:** Patient characteristics.

Year, Author	Patients No (BPD. vs. Control)	Birth Weight(gr) (BPD. vs. Control)	Nature of Birth (BPD. vs. Control)	Gestational Age at Delivery (BPD. vs. Control)	Apgar Score (Median) (BPD. vs. Control)	Antenatal Steroids (n) (BPD. vs. Control)	Surfactant Replacement Therapy(n) (BPD vs. Control)	Presence of Patent Ductus Arteriosus (PDA) (n) (BPD vs. Control)
2009, Joseph [14]	11 vs. 23	944 ± 337 vs. 1512 ± 314	N/A	27.4 ± 2.9 vs. 31.1 ± 1.4	A1:6.5 vs. 8 A5: 7 vs. 8	N/A	N/A	0
2015, Sellmer [15]	48 vs. 86	835 vs. 1170	CS: 31 vs. 56	26 vs. 29	A1: 7 vs. 8 A5: 10 vs. 10	33 vs. 82	30 vs. 31	31 vs. 29
2017, Montaner [16]	40 (BPD or death) vs. 77 (no BPD or death)	755 vs. 1040	N/A	26.6 vs. 28.1	N/A	33 vs. 64	39 vs. 53	39 vs. 38
2018, Blanco [17]	28 (BPD and/or death) vs. 82 (no BPD)	840 (690–1050) vs. 1280	N/A	26.7 vs. 30.4	A1: N/A A5: 7.5 vs. 9	21 vs. 57	22 vs. 41	27 vs. 32
2019, Khan [18]	44 (moderate to severe BPD) vs. 24 (no to mild BPD)	955 ± 249 vs. 1158 ± 230	CS: 36 vs. 15 Vaginal: 8 vs. 8	26 ± 2 vs. 28 ±1	A1: 3.5 vs. 5 A5: 6 vs. 7	27 vs. 15	1st dose: 44 vs. 20 2nd dose: 18 vs. 1	N/A
2019, Abad [19]	15 vs. 86	850 vs. 1200	CS: 12 vs. 72	27.27 ± 1.3 vs. 29.13 ± 1.79	A5: 7 vs. 8	14 vs. 72	N/A	7 vs. 14
2019, Zhou Lin [20]	48 (moderate to severe BPD) vs. 99 (no to mild BPD)	1110 ± 273 vs. 1398 ± 304	CS: 33 vs. 61	29^+1^ vs. 30	A1: 7 vs. 9 A5: 9 Vs. 10	N/A	42 vs. 86	34 vs. 34
2021, Potchiurko [21]	25 vs. 27	820 vs. 1200	CS: 5 vs. 17	27 vs. 30	A1: 4 vs. 6 A5: 5 vs. 6	15 vs. 21	24 vs. 19	17 vs. 5
2021, Song [22]	138 (moderate to severe BPD) vs. 418 (no to mild BPD)	1100 vs. 1250	CS: 100 vs. 331	29.1 vs. 30	A1: 7 vs. 8 A5: 8 vs. 9	133 vs. 170	324 vs. 452	65 vs. 112

Abbreviations: BPD, bronchopulmonary dysplasia; CS, caesarian section; N/A, Non-applicable.

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
