# Peer review of "N-Terminal Pro-B Type Natriuretic Peptide as a Predictive Biomarker of Bronchopulmonary Dysplasia or Death Due to Bronchopulmonary Dysplasia in Preterm Neonates: A Systematic Review and Meta-Analysis"

_jpm, 2023, doi:10.3390/jpm13091287_

Round 1
Reviewer 1 Report
Brain natriuretic peptide (BNP) was isolated originally from porcine brain extracts but was soon defined as a cardiac natriuretic hormone. Together with the highly homologous atrial natriuretic peptide, it forms a dual natriuretic peptide system of the heart. The main stimulus for proBNP synthesis and secretion from cardiac myocytes is myocyte stretch.
Brain natriuretic peptide (BNP) is a 32 amino acid cardiac natriuretic peptide hormone The human BNP gene is located on chromosome 1 and encodes the prohormone proBNP. The biologically active BNP and the remaining part of the prohormone, NT-proBNP (76 amino acids) can be measured by immunoassay in human blood. Cardiac myocytes constitute the major source of BNP related peptides.
Other neurohormones may stimulate cardiac BNP production in different cardiac cell types.
In contrast to atrial natriuretic peptides (ANP/NT-proANP), which originate mainly from atrial tissue, BNP related peptides are produced mainly from ventricular myocytes. Ventricular (NT-pro)BNP production is strongly upregulated in cardiac failure and locally in the area surrounding a myocardial infarction. In peripheral organs BNP binds to the natriuretic peptide receptor type A causing increased intracellular cGMP production. The biological effects include diuresis, vasodilatation, inhibition of renin and aldosterone production and of cardiac and vascular myocyte growth. In mice BNP gene knockout leads to cardiac fibrosis, gene over-expression to hypotension and bone malformations. BNP is cleared from plasma through binding to the natriuretic peptide clearance receptor type C, but it seems relatively resistant to proteolysis by neutral endopeptidase NEP 24.11.
Clearance mechanisms for NT-proBNP await further study. While the plasma concentration of NT-proBNP and BNP is approximately equal in normal controls, NT-proBNP plasma concentration is 2-10 times higher than BNP in patients with heart failure.
This relative change in peptide levels may be explained by shifts in cardiac secretion and/or clearance mechanisms.
While NT-proBNP and ANP are well studied in adolescents their significance in baybies especially in erly preterms is less clear. Pulmonary hypertension is a life-threatening disease especially in children. Diagnosis can be challenging in children; the current diagnostic options-right heart catheterization and echocardiography-are invasive and/or investigator-dependent procedures. • Biomarkers could be useful in this context because they are investigator independent and easy to obtain through blood samples. Brain natriuretic peptide (BNP) and its N-terminal cleavage product (NT-proBNP) seem to be the most promising. The value of these biomarkers in the diagnostic approach of PH has already been investigated in adults, with promising results. Pediatric studies are still scarce. What is new: • The levels of BNP and NT-proBNP in pediatric patients differ strongly between the different categories of PH. Within the same category, the levels are more or less equal. • The relative changes could render them a prognostic marker in the follow-up of a certain individual patient. At this moment there is not enough evidence to rely on BNP or NT-proBNP in clinical treatment of patients with PH.
Nine studies were selected for inclusion in the present meta-analysis.
The analysis of standardized mean differences indicated that neonates that developed BPD had significantly higher values of NT-proBNP compared to preterm controls. This seems to be the first systematic review in the current literature that addresses the association between Nt-proBNP levels after birth with the subsequent development of BPD in preterm neonates by including relevant published data.
The outcomes of this meta analysis highlighted that serum NT-proBNP levels display a promising diagnostic performance for the development of severe BPD or death in premature neonates . Despite the strong indications for the usefulness of NT proBNP, its diagnostic accuracy and clinically relevance must be more thoroughly examined with future prospective studies.
The connection between NT-proBNP and bronchopulmonary dysplasia has been the subject of research in recent years. Studies have shown that premature infants with bronchopulmonary dysplasia may have elevated levels of NT-proBNP. The exact mechanism for this association is not fully understood, but it is believed that the stress on the heart caused by the lung disease and the resulting pulmonary hypertension may lead to increased NT-proBNP release.
Measuring NT-proBNP levels in premature infants with bronchopulmonary dysplasia can provide valuable information about the cardiac function and may help in the management and monitoring of these infants' medical condition.
However, it's essential to interpret the results carefully and in conjunction with other clinical findings, as elevated NT-proBNP levels can also be present in other conditions, such as heart failure, kidney disease, or respiratory distress unrelated to BPD.
While pooled analyses of mean differences in serum NT-proBNP and AUC of NT-proBNP is well presented, no conclusion about pooled sensitivity and specificity of NT-proBNP levels can be drawn. Could this be presented as well based on available data?
The potential clinical benefit of stratification according NT-proBNP in terms of therapeutic measures should be explained in more detail in the introduction and /or discussion section.
Author Response
Brain natriuretic peptide (BNP) was isolated originally from porcine brain extracts but was soon defined as a cardiac natriuretic hormone. Together with the highly homologous atrial natriuretic peptide, it forms a dual natriuretic peptide system of the heart. The main stimulus for proBNP synthesis and secretion from cardiac myocytes is myocyte stretch.
Brain natriuretic peptide (BNP) is a 32 amino acid cardiac natriuretic peptide hormone The human BNP gene is located on chromosome 1 and encodes the prohormone proBNP. The biologically active BNP and the remaining part of the prohormone, NT-proBNP (76 amino acids) can be measured by immunoassay in human blood. Cardiac myocytes constitute the major source of BNP related peptides.
Other neurohormones may stimulate cardiac BNP production in different cardiac cell types.
In contrast to atrial natriuretic peptides (ANP/NT-proANP), which originate mainly from atrial tissue, BNP related peptides are produced mainly from ventricular myocytes. Ventricular (NT-pro)BNP production is strongly upregulated in cardiac failure and locally in the area surrounding a myocardial infarction. In peripheral organs BNP binds to the natriuretic peptide receptor type A causing increased intracellular cGMP production. The biological effects include diuresis, vasodilatation, inhibition of renin and aldosterone production and of cardiac and vascular myocyte growth. In mice BNP gene knockout leads to cardiac fibrosis, gene over-expression to hypotension and bone malformations. BNP is cleared from plasma through binding to the natriuretic peptide clearance receptor type C, but it seems relatively resistant to proteolysis by neutral endopeptidase NEP 24.11.
Clearance mechanisms for NT-proBNP await further study. While the plasma concentration of NT-proBNP and BNP is approximately equal in normal controls, NT-proBNP plasma concentration is 2-10 times higher than BNP in patients with heart failure.
This relative change in peptide levels may be explained by shifts in cardiac secretion and/or clearance mechanisms.
While NT-proBNP and ANP are well studied in adolescents their significance in baybies especially in erly preterms is less clear. Pulmonary hypertension is a life-threatening disease especially in children. Diagnosis can be challenging in children; the current diagnostic options-right heart catheterization and echocardiography-are invasive and/or investigator-dependent procedures. • Biomarkers could be useful in this context because they are investigator independent and easy to obtain through blood samples. Brain natriuretic peptide (BNP) and its N-terminal cleavage product (NT-proBNP) seem to be the most promising. The value of these biomarkers in the diagnostic approach of PH has already been investigated in adults, with promising results. Pediatric studies are still scarce. What is new: • The levels of BNP and NT-proBNP in pediatric patients differ strongly between the different categories of PH. Within the same category, the levels are more or less equal. • The relative changes could render them a prognostic marker in the follow-up of a certain individual patient. At this moment there is not enough evidence to rely on BNP or NT-proBNP in clinical treatment of patients with PH.
Nine studies were selected for inclusion in the present meta-analysis.
The analysis of standardized mean differences indicated that neonates that developed BPD had significantly higher values of NT-proBNP compared to preterm controls. This seems to be the first systematic review in the current literature that addresses the association between Nt-proBNP levels after birth with the subsequent development of BPD in preterm neonates by including relevant published data.
The outcomes of this meta analysis highlighted that serum NT-proBNP levels display a promising diagnostic performance for the development of severe BPD or death in premature neonates . Despite the strong indications for the usefulness of NT proBNP, its diagnostic accuracy and clinically relevance must be more thoroughly examined with future prospective studies.
The connection between NT-proBNP and bronchopulmonary dysplasia has been the subject of research in recent years. Studies have shown that premature infants with bronchopulmonary dysplasia may have elevated levels of NT-proBNP. The exact mechanism for this association is not fully understood, but it is believed that the stress on the heart caused by the lung disease and the resulting pulmonary hypertension may lead to increased NT-proBNP release.
Authors reply: in the present systematic review we underlined the importance of determining the diagnostic accuracy of NT-proBNP in diagnosing BPD, as to date, there are no relevant data on the sensitivity and/or specificity of the method. (Lines 394-397)
Reviewer 2 Report
Thank you to the authors for this important piece of work.
Please find below my comments which I would appreciate if you could comment on.
Abstract:
1. The predictive capacity of NT-proBNP is mentioned. Perhaps could consider being more specific and mention BPD and/or death is what is being investigated.
Methods:
1. Was any language restriction used during the search. Please comment.
There need to be more information on the presence of PDA. Since PDA is also associated with a rise in NT-proBNP, it is imperative that this confounder is appropriately investigated to tease out the real effects of BPD on NT-proBNP. I think the authors need to state the number of infants with PDA and rule out the effects of PDA on NT-proBNP if possible. If not, this should be clearly mentioned in the limitations.
Well written.
Author Response
Abstract:
- The predictive capacity of NT-proBNP is mentioned. Perhaps could consider being more specific and mention BPD and/or death is what is being investigated.
Authors reply: the investigated outcomes were mentioned in the present revision as well as the inability to detect data related to the diagnostic accuracy of the method. (Lines 19-20 & 31-32)
Methods:
- Was any language restriction used during the search. Please comment.
Authors reply: this point is clarified in the present revision (Lines 89-90)
There need to be more information on the presence of PDA. Since PDA is also associated with a rise in NT-proBNP, it is imperative that this confounder is appropriately investigated to tease out the real effects of BPD on NT-proBNP. I think the authors need to state the number of infants with PDA and rule out the effects of PDA on NT-proBNP if possible. If not, this should be clearly mentioned in the limitations.
Authors reply: In table 2 , a new column regarding PDA incidence between BPD subjects and controls was added. In lines 18-20 , 88-89 , 304-325 ,336 -339 ,403-407 we performed the changes indicated by the reviewers in our text.